# OpenReview forum: "On scalable oversight with weak LLMs judging strong LLMs"
_NeurIPS.cc/2024/Conference — NeurIPS 2024 poster_

### Official Review · Reviewer_MBPi · 2024-07-12

**Soundness:** 3
**Presentation:** 3
**Contribution:** 3
**Rating:** 7
**Confidence:** 4

**Summary:**

The paper provides a comprehensive study of scalable oversight across 3 access: (1) task, (2) scalable oversight protocol and (3) judge capacity / strength. The authors focus on inference-time scalable oversight, i.e. the debater models are not trained to do debate with a given judge. The authors consider several new tasks in the context of scalable oversight, such as multimodal and closed tasks (as opposed to extractive). They also consider novel protocols: open consultancy and open debates.

The results are overall quite mixed, with debate typically doing better than consultancy, but often not substantially outperforming direct QA outside of extractive tasks.

**Strengths:**

1. The study is carefully designed: the authors carefully vary the judge strength, tasks and scalable oversight protocols, and study the effect of each part.

2. The study is quite comprehensive, covering many judge models and tasks.

3. The presentation is balanced: the authors are not over-selling the results. Most of the observations are treated as weak evidence towards a certain hypothesis. The authors also clearly discuss limitations of the study.

4. The results on the debate outperforming consultancy are interesting, and provide some hope for debate as a scalable oversight protocol.

5. Using weak model judges as opposed to information asymmetry is in my opinion a very reasonable idea.

**Weaknesses:**

1. The debaters and the judges are all prompted models. These models are not trained to be particularly convincing to the judge, and the judge is not trained to be an accurate judge. The authors mention training the models as an interesting direction of future work.

2. The results are overall pretty mixed. For example, in Figure 1 on closed and multimodal tasks it appears that almost aways QA is better than both debate and consultancy. In other words, the judge can do better without any scalable oversight. Is that correct?

**Questions:**

1. I wonder if the reason for consultancy working worse than debate is sycophancy of the judge: when it gets only an argument for one side, it is inclined to follow that argument, because it's an LLM trained with RLHF. I wonder if this is not indicative of what a human judge would do.

2. From manual inspection, do you think the reason for poor results in closed and multimodal tasks is (a) poor debate / consultancy arguments or (2) bad decisions conditioned on those arguments?

**Limitations:**

Limitations are adequately addressed.

---

> ### Author Rebuttal · Authors · 2024-08-05
>
> Thank you for your thorough review, and interesting questions and comments. We are glad you found the study to be “carefully designed” and “comprehensive”, and appreciated the use of weak models as a complement to information asymmetry.
>
> Weaknesses:
>
> 1. **Training vs prompting:** we agree training would be great, the main reason we didn’t focus here is the extra cost involved. We instead prioritised a range of tasks which we think is the best decision as it showed task type has a big effect on performance. We are planning followup work on training.
> 2. **Direct QA vs debate:** Please see the top-level comment for our response.
>
> Questions
>
> 1. **Sycophancy:** This is an interesting hypothesis which could potentially be tested by conducting a human study (something we reference in future work), or by somehow trying to remove the judge errors due to sycophancy, perhaps by using base models for judges instead of RLHF’ed models (though this may induce other issues). Khan et al. have results for consultancy with human judges (on the QuALITY task) which show scores increase from a maximum of 60% with LLM to 80%  with humans suggesting LLM judges are significantly worse at this task. We suspect this is not necessarily just sycophancy though, it’s more likely that human judges ask better probing questions to the consultant than LLM judges do. We would be happy to add discussion of this to the revision.
> 2. **Poor debate vs. poor judging:** We think it is more likely bad judgments as the debate arguments seemed plausible/good, though this is from manual inspection (we ran a small human judging session as researchers). We think judges could do a lot better at deliberating based on the (conflicting) evidence in front of them. We think this could be improved via limited judge fine-tuning on the task of judging debates (something they haven’t been exposed to much in existing fine-tuning).

---

### Official Review · Reviewer_P56G · 2024-07-12

**Soundness:** 2
**Presentation:** 3
**Contribution:** 3
**Rating:** 5
**Confidence:** 3

**Summary:**

This paper focuses on scalable oversight protocols using debates between AI agents to align superhuman AI with human supervision. By studying debates judged by less capable LLMs across various tasks, the research finds that Debates, especially without artificial limitations on judges, more effectively bridge capability gaps compared to Consultancy methods. Stronger debaters also lead to higher judge accuracy, demonstrating the utility of the debate format for scalable oversight.

**Strengths:**

The study demonstrates the generalization of the debate protocol, which shows its applicability not only in extractive QA but also its superiority in other tasks compared to consultancy. It was found that debate can reduce the magnification of errors.

**Weaknesses:**

While I understand that this paper seeks to examine the effectiveness of Debate and Consultancy across various tasks, there are still some concerns:
1. This paper claims “Debate is likely more promising as a scalable oversight protocol than Consultancy”. However, some experiments indicate that while Debate yields better results than Consultancy, both perform worse compared to directly answering the question (as shown in QA without article in Figure 1).

2. Many results are without corresponding discussions or explanations.

i) Why LLMs using Debate and Consultancy would be worse in closed QA and multimodal tasks (Figure 1)?

ii) In Section 4.2, when using a weak judge with Open Debate protocol, this paper claims “weaker judges can struggle to discern that this is correct”. When judges get stronger, the judge accuracy improves. Does this imply that Debate may not be effective with weak judges? Additionally, as judges strengthen, how can we determine whether the improvement in accuracy is due to the Debate protocol or simply the enhanced capabilities of the judge?

iii) In Section 4.2, when the judge is weak with the Open Consultancy protocol, there is a similar phenomenon with ii), why?

**Questions:**

Please refer to Weaknesses.

**Limitations:**

The authors discuss the limitations and potential societal impact.

---

> ### Author Rebuttal · Authors · 2024-08-05
>
> We thank the reviewer for their insightful comments and agree that further discussions and explanations will help the reader better understand our results. We’d be happy to include these in a revision. Please see below for a more detailed response to your questions:
>
> i) **QA without article vs debate:** Please see the top-level comment for our response. This similarly applies to e.g. consultancy vs QA without article.
>
> ii)
> a) **Is debate ineffective with weak judges?** We think yes, that there is some evidence of a certain capability threshold below which judges are unable to effectively judge a debate. Evidence: Fig 1, extractive: the debate accuracy for Gemma7B is significantly lower than other judge models. Fig 1, closed: the debate accuracy for Gemma7B is significantly lower than other judge models, and additionally is significantly worse than QA without article. Fig 2: in open debate, accuracy of Gemma7B is significantly lower. In the current plot this is most evident in extractive tasks, but we have an updated version of Figure 2 (see top-level rebuttal for figure pdf) that also shows this for closed tasks (combined results rather than splitting by protagonist correct/incorrect). We will use this updated figure 2 and provide this discussion in the revision.
> b) **As judges' capabilities strengthen, how can we determine whether their improvement is due to the debate vs their overall capability strengthening?** We can look at the accuracy increase as the judge model strength increases for different protocols. For GPT3.5 increase over Gemma7B, debate the accuracy increases are: (extractive: 0.14, closed: 0.09) which are notably larger than the score increases for QA without article (extractive: 0.04, closed: 0.01) and QA with article (extractive: 0.02). We will add discussion of these score differences (and other accuracy increases between models) to the revision.
>
> iii) For **open consultancy**:
> a) **Is open consultancy ineffective with weak judges?** We see less evidence of a threshold here, we think because the weak judge can do quite well by deferring to the open consultant and this deference is not too difficult for Gemma7B.
> b) **As judges' capabilities strengthen, how can we determine whether their improvement is due to the open consultancy protocol vs their overall capability strengthening?** Similarly to ii), we calculate the open consultancy score increases for GPT3.5-Gemma7B as (extractive: 0.08, closed: 0.13). We will add discussion of these score differences (and other accuracy increases between models) to the revision.

---

> > ### Comment · Reviewer_P56G · 2024-08-13
> >
> > Thanks for your detailed response. It has addressed some of my concerns, I will raise the score to reflect this. However, I am still concerned that the performance and the corresponding analysis are limited in this paper. Besides, after reading the comments from other reviewers, it seems the novelty of this paper needs to be more clearly demonstrated. I will reduce my confidence as well.

---

### Official Review · Reviewer_HiQA · 2024-07-13

**Soundness:** 4
**Presentation:** 4
**Contribution:** 3
**Rating:** 8
**Confidence:** 2

**Summary:**

This paper is concerned with the study of scalable oversight methods, i.e, how can one devise methods that will allow humans to supervise and align superintelligent models (ASI) whose capacities (which include reasoning, strategic thinking, and deception) vastly exceed the ones of humans. Inspired both by recent works studying debate as a method for aligning strongly capable AI and by works modelling scalable oversight with smaller LLMs tasked to align stronger LLMs, the current work studies debate between more capable LLMs, as judged by a weaker LLM, both as a proxy for scalable oversight of ASI by humans and as a proxy for the richness of the signal the judge could provide to the strong LLMs during alignment training.

The authors back the task on extractive and closed QA tasks with 2 possible answers, where each of the 2 debating models are given a side and must persuade the judge to agree with them. This debate task is contrasted with the consultancy task, where a single model is given an arbitrary side and must persuade the judge to agree with it (the arguments for the opposing side are not visible). The study is large-scale, involving 9 tasks totaling 128 questions. A set of LLMs of varying sizes are used as judge to assess the effects of the gap between judge and debater capabilities. Open variants of debate and consultancy, where the judge/consultants are allowed to choose the answer they will argue for, are also investigated.

Among the important results of the paper, the authors find that under debate, for all judge sizes, the judges achieve better accuracy (predicting the correct answer) than under consultancy, highlighting the promise of debate as a promising alternative to RLHF as a basis for scalable oversight. Then, then show that in their setting, judges are convinced equally by consultants that have chosen the right versus the wrong answer, whereas in the debate case models that have chosen the right answer are believed more often, providing additional evidence for debate vs consultancy. Then, they show that judge accuracy increases as the capabilities of debaters increases (measured by their Elo in debates against other models), showing that debate scales with the capabilities of the LLMs to align. Their result also extends previous work on the debate task, with judges of the same strength as debaters, that was performed in a single task.

**Strengths:**

* The subject matter is important and of overarching importance to the Neurips community and beyond; the findings will be of particular interest to anyone concerned with AI safety.
* The paper is excellently written. The subject is not completely trivial and there are many setups and extensive experiments to present, but nevertheless the authors do an excellent job of explaining everything and putting it all together, highlighting the main results and the lessons learned as they go along as well as in the introduction and conclusion. The paper is very well contextualized in the related work and relations to prior art are precisely explained and motivate the current approach. I am not an expert in scalable oversight but I feel I have a much stronger grasp on the domain after reading the paper;
* The paper proposes to extend the study for a candidate for scalable oversight which follows naturally from previous work by casting it in a setting that is partially representative of the challenges ASI alignment poses. The authors study this as a scientific question, precisely reporting their findings and not overstating the extent to which debate is a definitive solution to scalable oversight. Extensive experiments support all of their claims and conclusions, and overall the paper (and its appendix) are information-rich.
* The authors highlight where their results agree or disagree with previous results in the literature.
* While debate as an alignment method is not novel it has not been studied with weaker judges judging stronger models, nor with such task variability (including knowledge-intensive tasks and reasoning-intensive tasks).
* All results come with clearly marked 95% CIs.

**Weaknesses:**

* One could have hoped that debate would perform better, compared to consultancy, but the gap between methods is still small (while consistent). How could one improve on debate to create a stronger signal for alignment? (this is hardly a weakness of the paper, however, but potential solutions to this could be discussed in the paper)
* Reproducibility is not perfect, since some results make use of chatgpt;

**Questions:**

* I like the result on chain of thought, it is counterintuitive and the explanation is plausible. Any idea on how to test this? (Maybe looking at attention matrices, or token influence?)
* line 275 I’m confused as to how models can both exhibit systematic positional bias and judge accuracy by unaffected by evaluation in both orders. How can this happen?
* line 344 “we don’t see such a clear trend of this advantage with increasing Elo” why do you think this is the case?
* (very minor) Summary sentences have too many commas, feels not that fluid (l353-357);

**Limitations:**

The main limitations of the work have been addressed by the authors at length in their paper, as far as I can tell. The expected societal impact of the work is likely to be overwhelmingly positive, as is usually the case with safety research. Maybe one note is that all alignment research is potentially misalignment research, in the wrong hands -- but this is hardly specific to this paper.

---

> ### Author Rebuttal · Authors · 2024-08-05
>
> We thank the reviewer for their detailed comments and are heartened to see that the reviewer appreciates the importance of the setting, likes the clarity of our exposition, and notes that our paper is a scientific study of protocols rather than promoting debate in particular.
>
> Weaknesses:
>
> * **Small gap between consultancy and debate:** we suspect the gap to increase when used as a training signal rather than inference protocol. We mention this in future work, but we’d be happy to directly reference the small gap as well and add more detail and potential solutions.
> * **Reproducibility:** While reproducibility is desirable, given that many of the most powerful models are closed, it would be a significant limitation if we were to avoid all closed models. To enhance reproducibility as much as possible, we evaluated on one strong open source model (Gemma).
>
> Questions:
>
> * **CoT:** one could look for substring match/rouge scores between judge CoT and debate reasoning. Token influence/attention could also be indicative, though perhaps more involved (and depending on the judge model may/may not be accessible). In principle, one could also ablate influence by inserting artificial reasoning errors into the debater arguments on some simple synthetic task, then comparing the influence on judge performance with and without CoT.
> * **Positional bias:** we have since found that positional bias reported in Khan et al (private communication) was due to them always setting the first answer as the correct answer. In our experiments we randomised the position of the correct answer. This then leads to the same mean positional bias as running both-orders (though both-orders has lower variance). We will update the text description to reflect this.
> * **Advantage with Elo:** we suspect the lack of trend is somewhat similar to Khan et al (on strong models, claude 2.1, gpt4t) which also didn’t see a strong trend (both on advantage and accuracy). Perhaps judge deliberation here is the bottleneck (and held constant) rather than debater skill (increasing). Through studying a more comprehensive range of tasks we were able to identify cases where previously reported trends for debate do and do not hold.
> * **Copy editing:** We’ll fix the commas in our revision.

---

> > ### Comment · Reviewer_HiQA · 2024-08-08
> >
> > Thank you for answering all of my questions! I am looking forward to follow-up work along the lines you mentioned. In light of the other reviews, and considering alignment is not my area of expertise I will lower my confidence score. I still think all my points stand and my grade is justified, and I would be very happy to see the paper accepted; however I acknowledge I am not familiar with all of the related work and thus encourage the AC to weigh other reviews more strongly than mine.

---

### Official Review · Reviewer_jsK8 · 2024-07-17

**Soundness:** 3
**Presentation:** 3
**Contribution:** 2
**Rating:** 4
**Confidence:** 3

**Summary:**

This paper primarily investigates scalable oversight by analyzing whether a weaker LLM can supervise a stronger LLM through various prompting pipelines.
Specifically, the paper compares the accuracy of responses from a weaker judge model under different interacting protocals with a stronger model, such as debate, consultancy, and direct question answering. The main finding is that having a strong model in debate, compared to consultancy, enables the weaker judge model to achieve better performance. The paper also provides a detailed analysis of different tasks, oversight protocols, and the capabilities of the judge models.

**Strengths:**

1. Compared to previous studies on debate and the judging/critique capabilities of models, this paper conducts more comprehensive experiments and ablations on the judge, primarily comparing the effects of different oversight protocols.
2. The presentation of the paper is clear and relatively easy to understand.

**Weaknesses:**

1. Lack of novelty: The comparison between consultancy and debate has already been explored in previous works [1, 2]. This paper essentially extends these comparisons to more tasks and analyses.
2. Lack of practical significance: Despite the extensive comparative analysis of judge protocols/models, there is no evident improvement brought by the weaker model to the stronger model. For instance, in the debate advocated by the paper, Figure 2 shows that even when the weaker judge model uses the strong model's debate as input, its performance does not surpass that of the strong model. Compared to previous work, I do not see how the paper's analysis provides substantial help in achieving effective scalable oversight.
3. The paper omits some highly relevant works analyzing the capabilities of LLM judges/critics, such as [3] [4].

---

[1] Debating with More Persuasive LLMs Leads to More Truthful Answers https://arxiv.org/pdf/2402.06782

[2] Debate Helps Supervise Unreliable Experts, https://arxiv.org/pdf/2311.08702

[3] Critique Ability of Large Language Models, https://arxiv.org/abs/2310.04815

[4] CriticBench: Benchmarking LLMs for Critique-Correct Reasoning, https://arxiv.org/abs/2402.14809

**Questions:**

See weaknesses.

**Limitations:**

See weaknesses.

---

> ### Author Rebuttal · Authors · 2024-08-05
>
> We thank the reviewer for their assessment. We emphasise that the key contribution of our paper is a rigorous, carefully-controlled scientific study of various scalable oversight protocols, rather than showing a particular protocol is better than others, see our top-level comment. We encourage the reviewer to consider our results in the context of the scalable oversight (see below for references on this area) setting with a weak judge overseeing a strong learner, in which the aim is to **improve the accuracy of the weak judge**. In particular, it is **not** our goal to improve the performance of the strong model.
>
> ## Weaknesses:
> 1. **Novelty**: Running similar experiments to [1,2] on a range of tasks has revealed some important differences to naively extrapolating the results on the single extractive task of [1,2]. The consultancy vs debate results of works [1,2] do in fact generalise to other tasks (see Introduction, bullets 1 and 5); however, QA without article vs. debate had different results on closed tasks, compared to what is found in the single extractive task of [1,2]. Our paper also has other novel findings (see Introduction, bullets 2, 3, 4) including previously untested open protocols. (Further, since submission, we have refined our analysis of section 4.3 on persuasiveness in debate, to split Elo calculations by task type, which shows the results of [1,2] on Elo scores also do not generalise to the closed tasks. We will update Figure 3 and its discussion accordingly, see top-level rebuttal for figure pdf. Our rebuttal does not rest on this amendment.) Overall, we think it is novel to extend analysis to new tasks/models/protocols, especially when doing so allowed us to show that the earlier work’s results do not generalise. [1] is an ICML 2024 best paper award winner and we think it is important that the community is aware of when its results do and do not apply.
> 2. **Practical significance**:
> - Improvement by the weaker model to the stronger model: this is not what we intend to show, instead we wanted to investigate what affects the weak judge’s accuracy when given access to protocols involving stronger models.
> - Weaker model does not surpass strong model performance: we do not aim for this, which would be a surprising finding (and outside of the scalable oversight setting). Figure 2 does show though, when the protagonist/consultant is incorrect (light cross) debate with weak judge has higher accuracy than in open consultancy, in which the weak judge often follows the strong consultant’s (incorrect) answer. The significance of this is that, for harder tasks, where the consultant would be incorrect more often, debate will highlight this to the weak judge more effectively. We have an updated version of Figure 2 (see top-level rebuttal for figure pdf) enabling easier comparison of accuracy when splitting by protagonist correct/incorrect. We will use this updated Figure 2 and provide this discussion in the revision.
> - It helps advance scalable oversight by providing evidence of the efficacy of proposed protocols. This does not rest on improving weaker models beyond the strong model’s performance (that would be an extreme standard to hold scalable oversight to, and is perhaps missing the point of scalable oversight research). The way we advance scalable oversight research is providing empirical evidence about the performance of the protocols, both positive and negative results are useful here.
> 3. **Citations**: We’d be happy to cite these additional works. How we differ: [3,4] don’t consider the setting of a weak judge overseeing a strong learner (which is our primary focus).
>
> ## References that study the same scalable oversight setting (weak supervisor, strong learner) as us:
>
> Theoretical
> - G. Irving, P. Christiano, and D. Amodei. AI safety via debate. arXiv preprint arXiv:1805.00899, 2018.
> - P. Christiano, B. Shlegeris, and D. Amodei. Supervising strong learners by amplifying weak experts. arXiv preprint arXiv:1810.08575, 2018.
> - J. Leike, D. Krueger, T. Everitt, M. Martic, V. Maini, and S. Legg. Scalable agent alignment via reward modeling: a research direction. arXiv preprint arXiv:1811.07871, 2018.
> - J. Brown-Cohen, G. Irving, and G. Piliouras. Scalable AI safety via doubly-efficient debate. ICML 2024 (Oral). https://openreview.net/forum?id=6jmdOTRMIO
>
> Empirical scalable oversight/debate
> - B. Barnes and P. Christiano. Writeup: Progress on AI Safety via Debate, 2020. URL https://www.alignmentforum.org/posts/Br4xDbYu4Frwrb64a/writeup-progress-on-ai-safety-via-debate-1
> - A. Parrish, H. Trivedi, N. Nangia, V. Padmakumar, J. Phang, A. S. Saimbhi, and S. R. Bowman. Two-turn debate doesn’t help humans answer hard reading comprehension questions. arXiv preprint arXiv:2210.10860, 2022.
> - J. Michael, S. Mahdi, D. Rein, J. Petty, J. Dirani, V. Padmakumar, and S. R. Bowman. Debate helps supervise unreliable experts. arXiv preprint arXiv:2311.08702, 2023.
> - A. Khan, J. Hughes, D. Valentine, L. Ruis, K. Sachan, A. Radhakrishnan, E. Grefenstette, S. R. Bowman, T. Rocktäschel, and E. Perez. Debating with more persuasive LLMs leads to more truthful answers. ICML, 2024 (Oral). https://openreview.net/forum?id=iLCZtl7FTa
> - A. Radhakrishnan. Anthropic fall 2023 debate progress update, 2023.
>
> Weak to strong generalisation
> - C. Burns, P. Izmailov, J. H. Kirchner, B. Baker, L. Gao, L. Aschenbrenner, Y. Chen, A. Ecoffet, M. Joglekar, J. Leike, et al. Weak-to-strong generalization: Eliciting strong capabilities with weak supervision. arXiv preprint arXiv:2312.09390, 2023.
>
> Sandwiching/scalable oversight setup
> - A. Cotra. The case for aligning narrowly superhuman models. In AI Alignment Forum, 2021.
> - S. R. Bowman, J. Hyun, E. Perez, E. Chen, C. Pettit, S. Heiner, K. Lukoši  ̄ut ̇e, A. Askell, A. Jones, A. Chen, et al. Measuring progress on scalable oversight for large language models. arXiv preprint arXiv:2211.03540, 2022.

---

> > ### Comment · Reviewer_jsK8 · 2024-08-13
> >
> > Thank you for the rebuttal, which has addressed some of my concerns.
> >
> > The authors clarified that the main contribution of the paper is "providing evidence of the efficacy of protocols," by `testing more tasks/models/protocols based on [1][2]`. Although the paper offers some mixed conclusions across different types of tasks (I acknowledge that these additional results may be valuable to researchers in specific sub-areas), it does not provide significantly more insights compared to previous work and is more like a replication report of [1][2]. Therefore, I will maintain my initial socre.
> >
> > Considering that the authors are more familiar with the scalable oversight setting, I have lowered my confidence and hope the Area Chair will consider the opinions of other reviewers more. However, I still believe this paper does not meet the NeurIPS standard and recommend submitting it to the *CL series instead.
> >
> > ---
> > [1] Debating with More Persuasive LLMs Leads to More Truthful Answers https://arxiv.org/pdf/2402.06782
> >
> > [2] Debate Helps Supervise Unreliable Experts, https://arxiv.org/pdf/2311.08702

---

### Author Rebuttal · Authors · 2024-08-05

We thank the reviewers for their thoughtful comments. We are glad to hear that reviewers identified our comprehensive experimentation and ablation, the importance of using weaker judge models to supervise stronger debaters, and the clarity of our writing and presentation as strengths of this work.

## QA without article vs debate
We address the following comment that came up in multiple reviews:

Reviews P56G, MBPi: **In the closed setting (where debaters don't have privileged information), debate often does not outperform QA without article.**

We would like to emphasise two points:
1. Our paper is a study comparing scalable oversight protocols, rather than a paper that introduces a particular method (eg debate) and shows it is better than all others (indeed reviewer HiQA notes the relative performance of debate is hardly a weakness of our paper). We are happy to add further emphasis on this in a revision.
2. In light of the prior point, we view this key finding as an interesting and perhaps surprising result of our paper. This is not what one would have expected from naively extrapolating the result of Khan et al. 2024, which only studied a single extractive task (QuALITY). This demonstrates an important contribution: evaluating on a wider range of tasks reveals different performances of the scalable oversight protocols.

As for the interpretation of this result: One hypothesis is that because our current judges are instruction-tuned models, trained for general purposes with supervised finetuning and RLHF, these models favor QA without article over debate: QA is typically the format of evaluation benchmarks which are used to select finetuning approaches, and which may be more common in the fine-tuning data (e.g. users typically ask questions and expect an answer). We suspect that judging a debate, in a discerning manner, is more out-of-distribution. Our results provide directions for future research: perhaps debaters which are fine-tuned to convince these judges will be able to learn to produce more in-distribution arguments for them, or perhaps judges can be fine-tuned to make more efficient use of the information they're presented with. It could also be interesting to test this by comparing to human debaters, who may have a more balanced distribution. We will add this discussion to the paper.

## Updates to figures 2 and 3

We attach (see pdf) updates to figures 2 and 3. Figure 2 now shows (top) combined results to better compare open debate vs open consultancy, and (bottom) separately shows accuracy split by correct/incorrect. Figure 3 now shows persuasiveness results split by task type (previously across all tasks) highlighting that the results of Khan et al., 2024 do not generalise to closed tasks. We reference these in rebuttals below.

---

### Decision · Program_Chairs · 2024-09-25

**Decision:**

Accept (poster)

**Comment:**

The paper presents a study of various protocols for evaluating strong LLMs by weaker evaluators. The evaluators/judges are intended to be humans in reality but represented by weaker LLMs in the submission, which allows this work to perform controlled experiments for various capability gaps between the judges and the LLMs being evaluated.

While some of the reviewers point out the paper's limited novelty in terms of offering new methodology, the metareviewer finds the paper's *intended* contribution -- a systematic assessment of various existing oversight protocols under more controlled conditions and using more reasonable judge models than has been done before, as well as empirical insights that stem from this assessment -- very valuable. The metareviewer expects this work to have notable impact on large model alignment research. Many of the questions that reviewers posed (see reviews *P56G*, *HiQA*, and *MBPi*) are great examples of scientific hypotheses that the research community is likely to start formulating based on this study. Testing them is likely to have tangible impact on LM training and deployment.

Under the assumption that the authors incorporate the rebuttal information that helped address the reviewers' concerns as well as the Figure 2 and 3 adjustments into the manuscript, the metareviewer recommends this paper for acceptance.